# Shape, Light, and Material Decomposition from Images using Monte Carlo Rendering and Denoising

**Jon Hasselgren**
NVIDIA

**Nikolai Hofmann**
NVIDIA

**Jacob Munkberg**
NVIDIA

## Abstract

Recent advances in differentiable rendering have enabled high-quality reconstruction of 3D scenes from multi-view images. Most methods rely on simple rendering algorithms: pre-filtered direct lighting or learned representations of irradiance. We show that a more realistic shading model, incorporating ray tracing and Monte Carlo integration, substantially improves decomposition into shape, materials & lighting. Unfortunately, Monte Carlo integration provides estimates with significant noise, even at large sample counts, which makes gradient-based inverse rendering very challenging. To address this, we incorporate multiple importance sampling and denoising in a novel inverse rendering pipeline. This improves convergence and enables gradient-based optimization at low sample counts. We present an efficient method to jointly reconstruct geometry (explicit triangle meshes), materials, and lighting, which substantially improves material and light separation compared to previous work. We argue that denoising can become an integral part of high quality inverse rendering pipelines.

## 1   Introduction

Differentiable rendering shows great promise for accurate multi-view 3D reconstruction from image observations. NeRF [39] use differentiable volume rendering to create high quality view interpolation through neural, density-based light-fields. Surface-based methods apply signed distance fields [43, 47, 68, 62, 73] or triangle meshes [41, 54] to capture high quality geometry. Recent work [6, 41, 75] further decompose these representations into geometry, material, and environment light.

Most aforementioned methods rely on appearance baked into neural light fields or apply simple shading models. Typically, direct lighting without shadows is considered, combined with pre-filtered representations of environment lighting [41, 73]. Some methods account for shadowing and indirect illumination [6, 75, 77], but often lock the geometry optimization when sampling the shadow term, or rely on learned representations of irradiance. While results are impressive, the deviations from physically-based shading models makes it harder for these methods to plausibly disentangle shape, material and lighting, as shown in Figure 1.

In theory, it is straightforward to replace the rendering engines of these 3D reconstruction methods with photorealistic differentiable renderers [31, 36, 46, 45, 71, 72] and optimize in a setting with more accurate light simulation, including global illumination effects. In practice, however, the noise in multi-bounce Monte Carlo rendering makes gradient-based optimization challenging. Very high sample counts are required, which result in intractable iteration times.

In this paper, we bridge the gap between current multi-view 3D reconstruction and physically-based differentiable rendering, and demonstrate high-quality reconstructions at competitive runtime performance. We attack the challenging case of extracting *explicit triangle meshes*, PBR materials

---

Project page: https://nvlabs.github.io/nvdiffrecmc/

36th Conference on Neural Information Processing Systems (NeurIPS 2022).

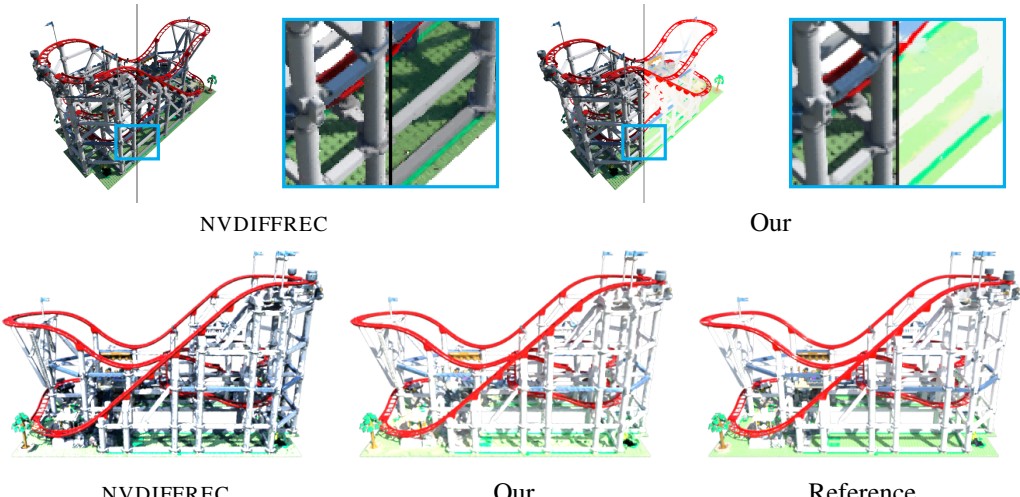

NVDIFFREC             Our

NVDIFFREC       Our       Reference

Figure 1: NVDIFFREC [41] successfully reconstructs complex geometry from multi-view images, but struggles with the material & light separation. In the top row, we visualize split-screens of the rendered reconstruction and the diffuse albedo texture. Note that NVDIFFREC bakes most of the lighting in the albedo texture, which hurts quality in relighting scenarios (shown in the bottom row). In contrast, by leveraging a more advanced renderer, we successfully disentangle material and lighting (note the lack of shading in the albedo texture), and improve relighting quality. The dataset consists of 200 views of the Rollercoaster from LDraw resources [29] (CC BY-2.0).

and environment lighting from a set of multi-view images, in a format directly compatible with current DCC tools and game engines. For improved visual fidelity, we compute direct illumination using Monte Carlo integration with ray tracing, and add several techniques to combat the increased noise levels. By carefully trading variance for bias, we enable efficient gradient-based optimization in a physically-based inverse rendering pipeline. Compared to previous 3D reconstruction methods, our formulation primarily improves material and light separation.

Concretely, we reduce variance by combining multiple importance sampling [58] and differentiable denoisers. We evaluate both neural denoisers [2, 10, 22] and cross-bilateral filters [52] in our pipeline. Furthermore, we decompose the rendering equation into albedo, demodulated diffuse lighting and specular lighting, which enable precise regularization to improve light and material separation.

## 2   Previous Work

**Neural methods for multi-view reconstruction**    These methods fall in two categories: *implicit* or *explicit* scene representations. NeRF [39] and follow-ups [38, 42, 49, 74, 64, 15, 40, 50, 69, 66], use volumetric representations and compute radiance by ray marching through a neurally encoded 5D light field. While achieving impressive results on novel view synthesis, geometric quality suffers from the ambiguity of volume rendering [74]. Surface-based rendering methods [43, 47, 68, 62] optimizing the underlying surface directly using implicit differentiation, or gradually morph from a volumetric representation into a surface representation. Methods with explicit representation estimate 3D meshes from images, where most approaches assume a given mesh topology [35, 11, 12], but recent work also include topology optimization [34, 14, 54, 41].

**BRDF and lighting estimation**    To estimate surface radiometric properties from images, previous work on BTF and SVBRDF estimation rely on special viewing configurations, lighting patterns or complex capturing setups [30, 16, 17, 18, 65, 5, 8, 53, 20]. Recent methods exploit neural networks to predict BRDFs from images [13, 19, 32, 33, 44, 37]. Differentiable rendering methods [35, 11, 76, 12, 21] learn to predict geometry, SVBRDF and, in some cases, lighting via photometric loss.

Most related to our work are neural 3D reconstruction methods with *intrinsic decomposition* of shape, materials, and lighting from images [6, 7, 41, 73, 75, 77]. Illumination is represented using

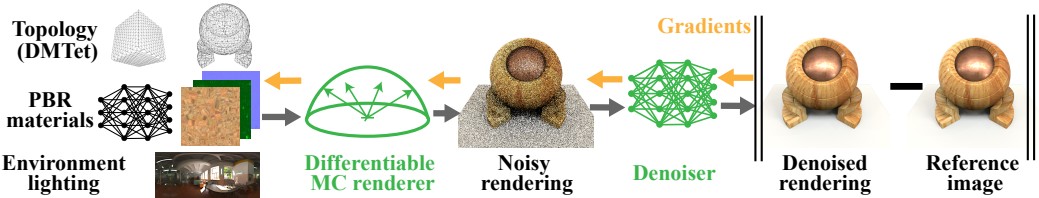

Figure 2: We extend NVDIFFREC [41] with a differentiable Monte Carlo renderer for direct illumination. Additionally, to reduce variance, we add a differentiable denoiser. These novel steps are highlighted in green. Following NVDIFFREC, the topology is parameterized using an SDF, and a triangular surface mesh is extracted in each iteration using DMTet [54], combined with spatially-varying PBR materials and HDR environment lighting. The system is supervised using only photometric loss on the rendered, denoised image compared to a reference, and gradients are back-propagated to the denoiser, shape, materials, and lighting parameters. All parameters are optimized jointly.

mixtures of spherical Gaussians [6, 41, 73, 77], pre-filtered approximations [7, 41], or low resolution environment maps [75]. When the shadowing term is accounted for [75], optimization is split into two passes where geometry is locked before the shadow term is sampled. Other approaches represent indirect illumination with neural networks [63, 77].

**Image denoisers**   Denoisers are essential tools in both real-time- and production renderers. Traditionally, variants of cross-bilateral filters are used [78], which require scene-specific manual adjustments. More recently, neural denoisers [2, 10, 22] trained on large datasets have shown impressive quality without the need for manual tuning, and are now incorporated in most production renderers. We directly incorporate differentiable versions of these denoisers in our pipeline. We are currently unaware of image denoisers applied in differentiable rendering, but we see a lot of potential for denoisers in physically-based inverse rendering going forward.

## 3   System

We target the challenging task of joint optimization of shape, material and environment lighting from a set of multi-view images with known foreground segmentation masks and camera poses. Our goal is to use physically-based rendering techniques to improve the intrinsic decomposition of lighting and materials, producing assets that can be relit, edited, animated, or used in simulation. As a proof-of-concept, we extend a recent approach, NVDIFFREC [41], which targets the same optimization task (shape, materials and environment lighting). Notably, they directly optimize a *triangular* 3D model, which has obvious benefits: it is easy to import and modify the reconstructed models in existing DCC tools, and a triangular representation can exploit hardware-accelerated differentiable rasterization [28]. In our setting, triangular 3D models also means we can leverage hardware-accelerated ray-tracing for efficient shadow tests. NVDIFFREC reports competitive results on view interpolation, material reconstruction, and relighting, and we will use their pipeline as a baseline in our evaluations.

Our system is summarized in Figure 2. A triangular mesh with arbitrary topology is optimized from a set of images through 2D supervision. Geometry is represented by a signed distance field defined on a three-dimensional grid and reduced to a triangular surface mesh through deep marching tetrahedra (DMTet) [54]. Next, the extracted surface mesh is rendered in a differentiable renderer, using the physically-based (PBR) material model from Disney [9]. This material model combines a diffuse term with an isotropic, specular GGX lobe [61]. A tangent space normal map is also included to capture high frequency shading detail. Finally, the rendered image is evaluated against a reference image using a photometric loss. In contrast to NVDIFFREC, which uses a simple renderer with deferred shading and the split-sum approximation for direct lighting (without shadows), we instead leverage a renderer which evaluates the direct lighting integral using Monte Carlo integration and ray tracing (shadow rays). We represent the scene lighting using a high dynamic range light probe stored as a floating point texture, typically at a resolution of $256 \times 256$ texels. Finally, to combat the inherent variance that comes with Monte Carlo integration, we leverage differentiable image denoising and multiple importance sampling.

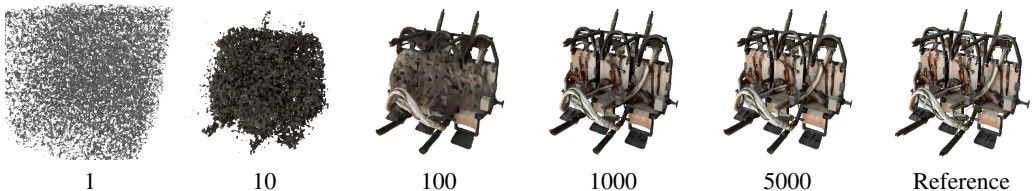

| 1 | 10 | 100 | 1000 | 5000 | Reference |

Figure 3: Visualization of the optimization process. Note that the initial guess for topology are randomized SDF values on the grid. After 1000 iterations, we already have a high quality topology and plausible materials and lighting for this complicated asset. Synthetic dataset with 200 frames, generated from a part of the Apollo capsule, courtesy of the Smithsonian [55] (CC0-1.0).

**Optimization task**  Let $\phi$ denote the optimization parameters (shape, spatially varying materials and light probe). For a given camera pose, $c$, our differentiable renderer produces an image $I_\phi(c)$. Given that we use Monte Carlo integration during rendering, this image inherently includes noise, and we apply a differentiable image denoiser, $D_\theta$, with parameters, $\theta$, to reduce the variance, $I_\phi^{\mathrm{denoised}}(c) = D_\theta(I_\phi(c))$. The reference image $I_{\mathrm{ref}}(c)$ is a view from the same camera. Given a photometric loss function $L$, we minimize the empirical risk

$$\operatorname*{argmin}_{\phi,\theta} \mathbb{E}_c \big[ L\big(D_\theta(I_\phi(c)), I_{\mathrm{ref}}(c)\big)\big] \tag{1}$$

using Adam [26] based on gradients w.r.t. the optimization parameters, $\partial L/\partial \phi$, and $\partial L/\partial \theta$, which are obtained through differentiable rendering. We use the same loss function as NVDIFFREC. An example of the optimization process is illustrated in Figure 3.

## 3.1  Direct Illumination

The outgoing radiance $L(\omega_o)$ in direction $\omega_o$ can be expressed using the rendering equation [24] as:

$$L(\omega_o) = \int_\Omega L_i(\omega_i) f(\omega_i, \omega_o)(\omega_i \cdot \mathbf{n}) d\omega_i. \tag{2}$$

This is an integral of the product of the incident radiance, $L_i(\omega_i)$ from direction $\omega_i$ and the BSDF $f(\omega_i, \omega_o)$. The integration domain is the hemisphere $\Omega$ around the surface normal, $\mathbf{n}$.

Spherical Harmonics (SH) [11] or Spherical Gaussians (SG) [6, 73] are often used as efficient approximations of direct illumination, but only work well for low- to medium-frequency lighting. In contrast, the *split sum* approximation [25, 41] captures all-frequency image based lighting, but does not incorporate shadows. Our goal is all-frequency lighting *including* shadows, which we tackle by evaluating the rendering equation using Monte Carlo integration:

$$L(\omega_o) \approx \frac{1}{N} \sum_{i=1}^N \frac{L_i(\omega_i) f(\omega_i, \omega_o)(\omega_i \cdot \mathbf{n})}{p(\omega_i)}, \tag{3}$$

with samples drawn from some distribution $p(\omega_i)$. Note that $L_i(\omega_i)$ includes a visibility test, which can be evaluated by tracing a shadow ray in direction $\omega_i$. Unfortunately, the variance levels in Monte Carlo integration with low number of samples makes gradient-based optimization hard, particularly with complex lighting. In Section 4 we propose several variance reduction techniques, which enable an inverse rendering pipeline that efficiently reconstructs complex geometry, a wide range of lighting conditions and spatially-varying BSDFs.

**Shadow gradients**  In single view optimization [60], shape from shadows [57], or direct optimization of the position/direction of analytical light sources, shadow ray visibility gradients [36, 3] are highly beneficial. However, in our multi-view setting (50+ views), similar to Loubet et al. [36], we observed that gradients of diffuse scattering are negligible compared to the gradients of primary visibility. Hence, for performance reasons, in the experiment presented in this paper, the shadow ray visibility gradients are detached when evaluating the hemisphere integral, and shape optimization is driven by primary visibility gradients, obtained from `nvdiffrast` [28].

# 4   Variance Reduction

We evaluate direct illumination with high frequency environment map lighting combined with a wide range of materials (diffuse, dielectrics, and metals). Strong directional sunlight, highly specular, mirror-like materials, and the visibility component can all introduce significant levels of noise. To enable optimization in an inverse rendering setting at practical sample counts, we carefully sample each of these contributing factors to obtain a signal with low variance. Below, we describe how we combat noise by using multiple importance sampling and denoising.

## 4.1   Multiple Importance Sampling

We leverage *multiple importance sampling* [58] (MIS), a framework to weigh a set of different sampling techniques to reduce variance in Monte Carlo integration. Given a set of sampling techniques, each with a sampling distribution $p_i$, the Monte Carlo estimator for an integral $\int_\Omega g(x)dx$ given by MIS is

$$\sum_{i=1}^{n} \frac{1}{n_i} \sum_{j=1}^{n_i} w_i(X_{i,j}) \frac{g(X_{i,j})}{p_i(X_{i,j})}, \quad w_i(x) = \frac{n_i p_i(x)}{\sum_k n_k p_k(x)}. \tag{4}$$

The weighting functions $w_i(x)$ are chosen using the *balance heuristic*. Please refer to Veach's thesis [58] or the excellent PBRT book [48] for further details.

In our case, we apply MIS with three sampling techniques: light importance sampling, $p_{\text{light}}(\omega)$, using a piecewise-constant 2D distribution sampling technique [48], cosine sampling, $p_{\text{diffuse}}(\omega)$, for the diffuse lobe, and GGX importance sampling [23], $p_{\text{specular}}(\omega)$, for the specular lobe. Unlike in forward rendering with known materials and lights, our material and light parameters are optimization variables. Thus, the sampling distributions, $p_i$, are recomputed in each optimization iteration. Following the taxonomy of differentiable Monte Carlo estimators of Zeltner et al. [71], our importance sampling is *detached*, i.e., we do not back-propagate gradients to the scene parameters in the sampling step, only in the material evaluation. Please refer to Zeltner et al. for a careful analysis of Monte Carlo estimators for differentiable light transport.

MIS is unbiased, but chaining multiple iterations of our algorithm exhibits *bias*, as the current, importance sampled, iteration dictates the sampling distributions used in the next iteration. Light probe intensities, for example, are optimized based on an importance sampled image, which are then used to construct the sampling distribution for the subsequent pass. For unbiased rendering, the sampling distribution must be estimated using a second set of uncorrelated samples instead. Furthermore, we explicitly re-use the random seed from the forward pass during gradient backpropagation to scatter gradients to the exact same set of parameters that contributed to the forward rendering. This approach is clearly biased [59], but we empirically note that this is very effective in reducing variance, and in our setting this variance-bias trade-off works in our favor.

## 4.2   Denoising

For differentiable rendering, the benefits of denoising are twofold. First, it improves the image quality of the rendering in the forward pass, reducing the optimization error and the gradient noise introduced in the image loss. Second, as gradients back-propagate through the denoiser's spatial filter kernel, gradient sharing between neighboring pixels is enabled. To see this, let's consider a simple denoiser, $O = X \circledast F$, where the noisy rendered image $X$ is filtered by a low-pass filter, $F$ ($\circledast$ represents an image space 2D convolution). Given a loss gradient, $\frac{\partial L}{\partial O}$, the gradients propagated back to the renderer are $\frac{\partial L}{\partial X} = \frac{\partial L}{\partial O} \circledast F^T$, which applies the same low-pass filter in case the filter is rotationally symmetric ($F = F^T$). In other words, the renderer sees *filtered* loss gradients in a local spatial footprint. While denoisers inherently trade a reduction in variance for increased bias, we empirically note that denoising significantly helps convergence at lower sample counts, and help to reconstruct higher frequency environment lighting. We show an illustrative example in Figure 9.

Following previous work in denoising for production rendering [2], Figure 4 shows how we separate lighting into diffuse, $\mathbf{c}_d$, and specular, $\mathbf{c}_s$ terms. This lets us denoise each term separately, creating denoised buffers, $D_\theta(\mathbf{c}_d)$, and $D_\theta(\mathbf{c}_s)$. More importantly, we can use *demodulated* diffuse lighting, which means that the lighting term has not yet been multiplied by material diffuse albedo, $\mathbf{k}_d$. In forward rendering, this is important as it decorrelates the noisy lighting from material textures, thus

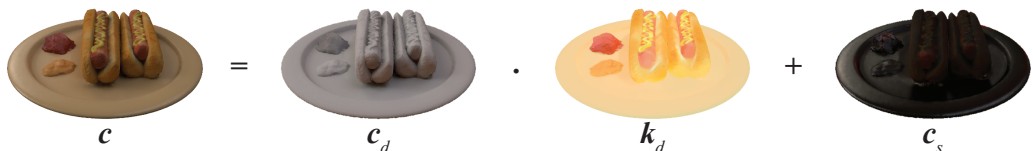

Figure 4: We separate lighting into diffuse lighting, $\mathbf{c}_d$, diffuse reflectance, $\mathbf{k}_d$, and specular lighting, $\mathbf{c}_s$. This enables fine-grained regularization and denoising without smearing texture detail.

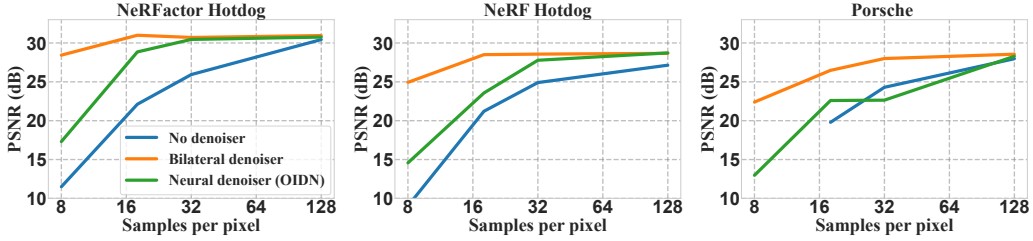

Figure 5: Ablation study on the effect of using different denoising algorithms during optimization at low sample counts on three different scenes of increasing complexity (from left to right). We plot averaged PSNR scores over 200 novel views, rendered without denoising, using high sample counts. In this experiment, we used decorrelated samples in the backward pass to highlight the effect of denoising. The most complex scene (Porsche) failed to converge at 8 spp without denoising.

selectively denoising the noisy Monte-Carlo estimates and avoiding to low-pass filter high-frequent texture information. In inverse rendering, we can additionally use it to improve material and light decomposition by adding regularization on the lighting terms, as disucssed in Section 5. We compose the final image as $\mathbf{c} = \mathbf{k}_d \cdot D_\theta(\mathbf{c}_d) + D_\theta(\mathbf{c}_s)$. We currently do not demodulate specular lighting because of the view dependent Fresnel term, but expect this to be improved in future work.

**Cross-bilateral filters** Cross-bilateral filters are commonly used to remove noise in rendered images [78]. To evaluate this family of denoisers in our inverse pipeline, we adapted Spatio-temporal Variance-Guided Filtering [52] (SVGF), which is a popular cross-bilateral filter using surface normals and per-pixel depth as edge-stopping guides. The denoiser is applied to *demodulated* diffuse lighting, to avoid smearing texture detail. We disabled the temporal component of SVGF, as we focus on single frame rendering, and implemented the filter as a differentiable module to allow for loss gradients to propagate back to our scene parameters.

**Neural denoisers** As a representative neural denoiser, we deploy the Open Image Denoiser (OIDN) [1], which is a U-Net [51] pre-trained on a large corpus of rendered images. The denoiser is applied to the rendered image before computing the image space loss. As the network model is fully convolutional, it is trivially differentiable, and we can propagate gradients from the image space loss, through the denoiser back to the renderer.

In Figure 5, we provide an overview on the effect of denoisers during 3D scene reconstruction. We observe that denoising is especially helpful at low sample counts, where we obtain similar reconstruction results at 8 spp with denoising, compared to 32 spp without a denoiser. At higher sample counts, however, the benefit from denoising diminishes, as variance in the Monte-Carlo estimates decreases. Given that denoisers enable significantly faster iteration times, we consider it a valuable tool for saving computational resources when fine-tuning model parameters for subsequent runs with high sample counts. Additionally, we empirically found that using a denoiser during reconstruction yields higher quality light probes, as can be seen in Figure 9, both at low and high sample counts.

We can also jointly optimize the denoiser parameters, $\theta$, with the 3D scene reconstruction task. i.e., the denoiser is *fine-tuned* for the current scene. Unfortunately, this approach has undesirable side-effects: Features tend to get baked into the denoiser network weights instead of the materials or light probe. This is especially apparent with the OIDN [1] denoiser, which produced color shifts

due to lack of regularization on the output. We got notably better results with the hierarchical kernel prediction architecture from Hasselgren et al. [22], which is more constrained. However, the results still lagged behind denoisers with locked weights. We refer to the supplemental material for details.

## 5 Priors

In our setting: 3D reconstruction from multi-view images with constant lighting, regularization is essential in order to disentangle lighting and materials. Following previous work [75, 41], we apply smoothness priors for albedo, specular, and normal map textures. Taking the albedo as an example, if $k_d(\mathbf{x})$ denotes the diffuse albedo at world space position, $\mathbf{x}$, and $\epsilon$ is a small random displacement vector, we define the smoothness prior for albedo as:

$$L_{\boldsymbol{k}_d} = \sum_{\mathbf{x}_{\text{surf}}} |\boldsymbol{k}_d(\mathbf{x}_{\text{surf}}) - \boldsymbol{k}_d(\mathbf{x}_{\text{surf}} + \epsilon)|, \tag{5}$$

where $\mathbf{x}_{\text{surf}}$ are the world space positions at the primary hit point on the object. We note that the smoothness prior is not sufficient to disentangle material parameters and light, especially for scenes with high frequency lighting and sharp shadows. Optimization tends to bake shadows into the albedo texture (easy) rather than reconstruct a high intensity, small area in the environment map (hard). To enable high quality relighting, we explicitly want to enforce shading detail represented by lighting, and only bake remaining details into the material textures. We propose a novel regularizer term that is surprisingly effective. We compute a monochrome image loss between the demodulated lighting terms and the reference image:

$$L_{\text{light}} = |\mathrm{Y}(\mathbf{c}_d + \mathbf{c}_s) - \mathrm{V}(I_{\text{ref}})|. \tag{6}$$

Here, $Y(\mathbf{x}) = (\mathbf{x}_r + \mathbf{x}_g + \mathbf{x}_b)/3$ is a simple luminance operator, and $V(\mathbf{x}) = \max(\mathbf{x}_r, \mathbf{x}_g, \mathbf{x}_b)$ is the HSV value component. The rationale for using HSV-value for the reference image is that the $\max$ operation approximates demodulation, e.g., a red and white pixel have identical values. We assume that the demodulated lighting is mostly monochrome, in which case $Y(\mathbf{x}) \sim V(\mathbf{x})$, and given that we need to propagate gradients to $\mathbf{c}_d$ and $\mathbf{c}_s$, $Y$ avoids discontinuities. This regularizer is limited by our inability to demodulate the reference image. The HSV-value ignores chrominance, but we cannot separate a shadow from a darker material. This has not been a problem in our tests, but could interfere with optimization if the regularizer is given too much weight. Please refer to the supplemental materials for complete regularizer details.

## 6 Experiments

In our experiments, we use NVDIFFREC [41] as a baseline, and refer to their work for thorough comparisons against related work. We focus the evaluation on the quality of material and light separation. At test time, all view interpolation results are generated without denoising at 2k spp. All relighting results are rendered in Blender Cycles at 64 spp with denoising [1]. Table 1 shows a quantitative comparison with NVDIFFREC and NeRFactor [75] on the NeRFactor relighting setup. Note that there is an indeterminate scale factor between material reflectance (e.g., albedo) and the

Table 1: Summarized relighting results for NeRFactor (CC-BY-3.0), NeRF (CC-BY-3.0) and our synthetic datasets. The NeRFactor dataset contains four scenes, each scene has eight validation views and eight different light probes (256 validation images). For the NeRF dataset (which contain higher frequency lighting), we use the Chair, Hotdog, Lego, Materials and Mic scenes, with eight validation views and four light probe configurations (160 validation images). Our dataset contains a variation of high and low frequency lighting with geometrically complex objects. The image metric scores are arithmetic means over all images.

| | NeRFactor synthetic | | | Nerf synthetic | | | Our synthetic | | |
| | PSNR↑ | SSIM↑ | LPIPS↓ | PSNR↑ | SSIM↑ | LPIPS↓ | PSNR↑ | SSIM↑ | LPIPS↓ |
|---|---|---|---|---|---|---|---|---|---|
| Our | 26.0 dB | 0.924 | 0.060 | 26.5 dB | 0.932 | 0.055 | 27.1 dB | 0.950 | 0.027 |
| NVDIFFREC | 24.8 dB | 0.910 | 0.063 | 23.3 dB | 0.889 | 0.076 | 23.7 dB | 0.925 | 0.049 |
| NERFACTOR | 22.2 dB | 0.896 | 0.087 | - | - | - | - | - | - |

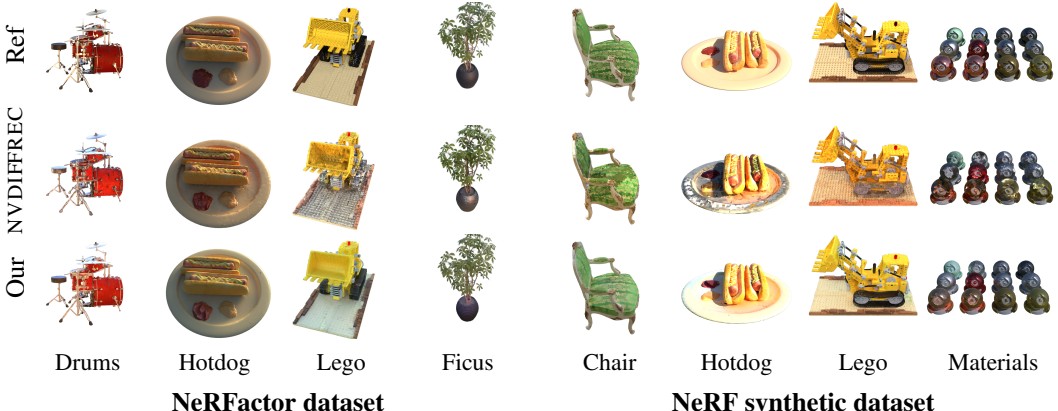

Figure 6: Relighting examples from the NeRFactor and NeRF synthetic datasets. The NeRF dataset contains high frequency lighting and global illumination, and is substantially more challenging than the NeRFactor version, which uses downsampled probes. Our results contain visible artifacts, but outperform the material separation of previous work.

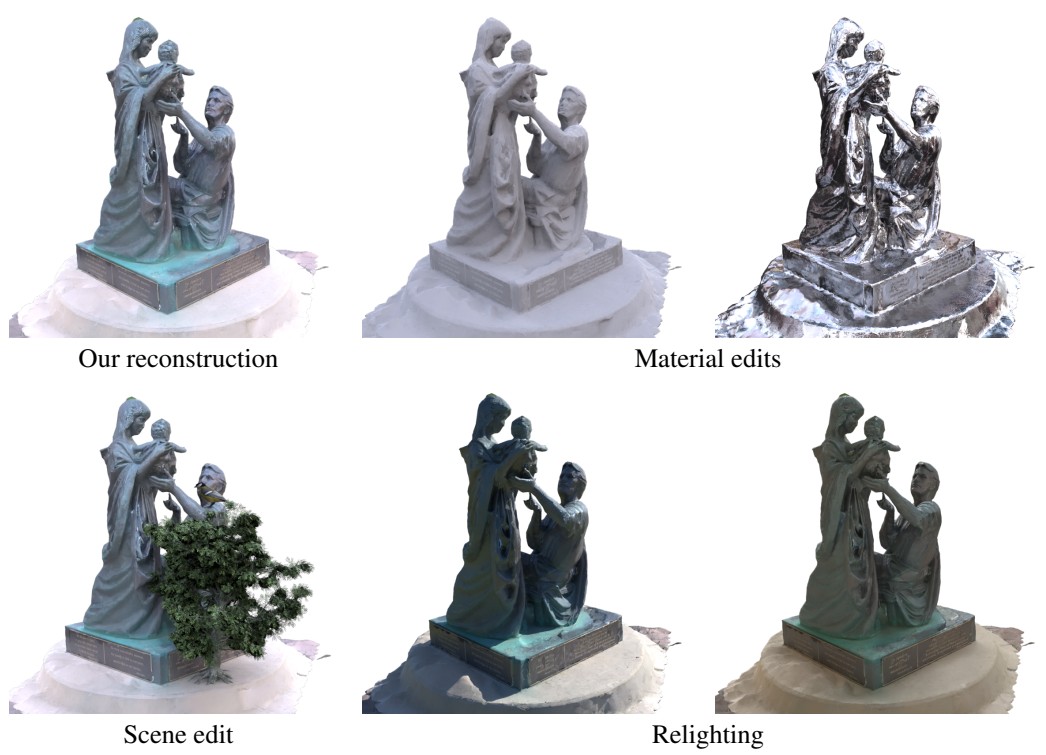

Figure 7: Manipulations of our extracted 3D model of the Family dataset in Blender. This scene is part of the Tanks&Temples [27] dataset (CC BY-NC-SA 3.0). Tree and bird models from TurboSquid.

light intensity. This is accounted for by scaling each image to match the average luminance of the reference for the corresponding scene. The same methodology is applied for all algorithms in our comparisons. We outperform previous work, providing better material reconstruction. Figure 6 shows visual examples.

We additionally perform relighting on the synthetic NeRF dataset, which is substantially more challenging than the NeRFactor variant, due to high frequency lighting and global illumination effects. NVDIFFREC produces severe artifacts, as exemplified by the Hotdog scene in Figure 6. Table 1 shows a significant increase in image quality for our approach. The visual examples show that our

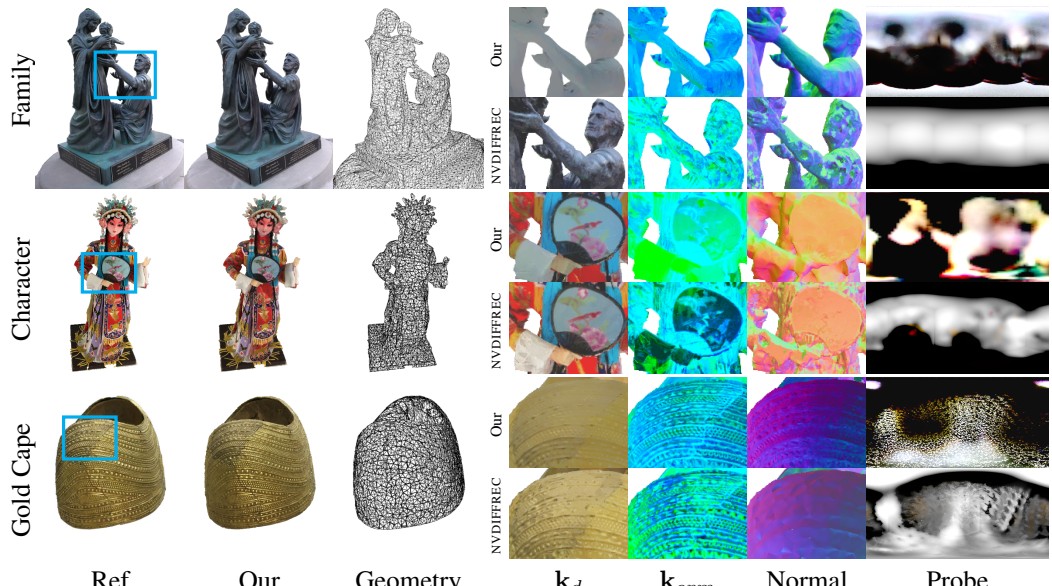

Figure 8: We show explicit decomposition of shape, materials and lighting, directly from photos with known poses. Character is part of the BlendedMVS [67] dataset (CC BY-4.0) and Gold Cape is part of the NeRD [6] dataset (CC BY-NC-SA 4.0).

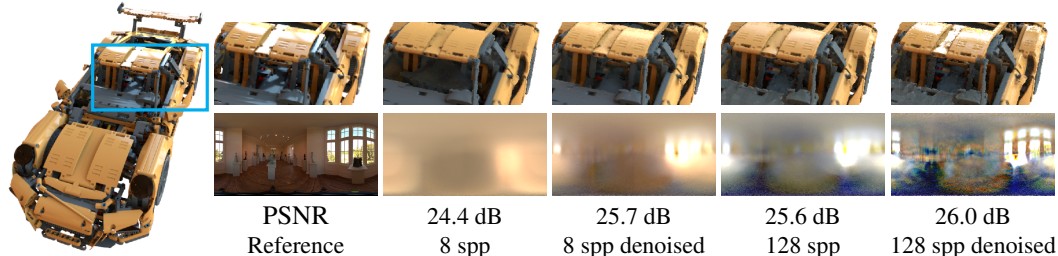

Figure 9: We show the benefits of denoising on the Porsche scene from LDraw resources [29] (CC BY-2.0). At low sample counts, denoising helps both with geometric reconstruction (in the cockpit) and to capture specular highlights. Even at 128 spp, denoising improves specular highlight and high frequency lighting details.

results are plausible, though not without artifacts. Finally, we constructed a novel synthetic dataset with three scenes with highly complex geometry to stress-test the system. Each scene contains 200 training views and 200 novel views for evaluation. The three scenes are shown in Figures 1 , 3 ,and 9. Quantitatively we outperform previous work by a larger margin, and Figure 1 shows very little shading in the albedo textures.

In Figures 7 and 8, we apply our method to datasets with real photos. These sets are more difficult, due to inaccuracies in foreground segmentation masks and camera poses. We extract triangle meshes that can be trivially edited in 3D modeling software, and in Figure 7 we use Blender to perform scene editing, material editing, and relighting. Note that the results look plausible with the inserted object being properly shaded and casting shadows on the statue, material editing works well with the learned environment lighting and relighting interacts properly with the learned materials. Figure 8 shows a breakdown of geometry, material parameters and environment light.

To study the impact of denoising, we optimized the Porsche scene w/ and w/o denoising. As shown in Figure 9, denoising improves both visual quality and environment lighting detail at equal sample counts. The noise levels varies throughout the scene, and we note that denoising is particularly helpful in regions with complex lighting or occlusion, such as the specular highlight and cockpit.

Neural light-fields, e.g. Mip-NeRF [4] excel at view interpolation. We enforce material/light separation through additional regularization, which slightly degrades view interpolation results, as

Table 2: View interpolation results. For reference, the NERFACTOR scores are 26.9 dB PSNR and SSIM of 0.930 on the NerFactor synthetic dataset, and Mip-NeRF has 35.0 dB PSNR and SSIM 0.978 on the Nerf synthetic dataset. The image metric scores are arithmetic means over all test images.

| | NeRFactor synthetic | | Nerf synthetic | | Our synthetic | | Real-world | |
| | PSNR↑ | SSIM↑ | PSNR↑ | SSIM↑ | PSNR↑ | SSIM↑ | PSNR↑ | SSIM↑ |
|---|---|---|---|---|---|---|---|---|
| Our | 29.6 dB | 0.951 | 28.4 dB | 0.938 | 25.6 dB | 0.934 | 25.29 dB | 0.899 |
| NVDIFFREC | 31.7 dB | 0.967 | 30.4 dB | 0.958 | 25.8 dB | 0.944 | 26.58 dB | 0.918 |

shown in Table 2. Our scores are slightly below NVDIFFREC, but, as shown above, we provide considerably better material and lighting separation.

**Compute resources** Tracing rays to evaluate the direct illumination is considerably more expensive than pre-filtered environment light approaches. We leverage hardware-accelerated ray intersections, but note that our implementation is far from fully optimized. Our method scales linearly with sample count, which gives us a simple way to trade quality for performance. With a batch size 8 at a rendering resolution of 512×512, we get the following iteration times on a single NVIDIA A6000 GPU.

| | NVDIFFREC | Our 2 spp | Our 8 spp | Our 32 spp | Our 128 spp | Our 288 spp |
|---|---|---|---|---|---|---|
| Iteration time | 340 ms | 280 ms | 285 ms | 300 ms | 360 ms | 450 ms |

Our denoising strategies enable optimization at low sample counts. Unless otherwise mentioned, for the results presented in the paper, we use high quality settings of 128+ rays per pixel with 5000×2 iterations (second pass with fixed topology and 2D textures), which takes ∼4 hours (A6000).

## 7 Conclusions

Our restriction to direct illumination shows up in scenes with global illumination. Similarly, we do not handle specular chains (geometry seen through a glass window). For this, we need to integrate multi-bounce path tracing, which is a clear avenue for future work, but comes with additional challenges in increased noise-levels, visibility gradients through specular chains, and drastically increased iteration times. Our renderer is intentionally biased to improve optimization times, but unbiased rendering could expect to generate better results for very high sample counts. Other limitations include lack of efficient regularization of material specular parameters and reliance on a foreground segmentation mask. Our approach is computationally intense, requiring a high-end GPU for optimization runs.

To summarize, we have shown that differentiable Monte-Carlo rendering combined with variance-reduction techniques is practical and applicable to multi-view 3D object reconstruction of explicit triangular 3D models. Our physically-based renderer clearly improves material and light reconstruction over previous work. By leveraging hardware accelerated ray-tracing and differentiable image denoisers, we remain competitive to previous work in terms of optimization time.

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
