# OpenReview forum: "Shape, Light, and Material Decomposition from Images using Monte Carlo Rendering and Denoising"
_NeurIPS.cc/2022/Conference — NeurIPS 2022 Accept_

### Official Review · Reviewer_1u76 · 2022-07-09

**Rating:** 5
**Confidence:** 3
**Soundness:** 3 good
**Presentation:** 3 good
**Contribution:** 3 good

**Summary:**

The paper proposes an inverse rendering pipeline for recovering geometry, reflectance and environment map from a set of multi-view images. The key contributions is a denoising step that significantly reduces variance of Monte Carlo light integration at lower sampling rates. The method achieved superior results compared to a direct lighting baseline (nvdiffrec) and Nerfactor.

**Questions:**

I wonder to what extent the proposed method could be used to handle inter-reflections and light refraction in e.g. translucent objects. To me this seems to be one of the biggest advantage of Monte Carlo integration versus SH/SG but I could not find relevant discussions in paper.

**Limitations:**

Yes limitations have been well discussed.

**Strengths And Weaknesses:**

Strengths:

- The idea to denoise Monte Carlo rendering is well motivated and well implemented. To the best of my knowledge this idea is original.
- Results are convincing. Particularly the comparison with nvdiffrec baseline shows considerable improvement in novel rendering and relighting. Compared to SG and SH representations, the recovered environment map contains high fidelity details, evident in Figure 2 in the appendix.
- Limitations have been discussed in detail. I appreciate authors honesty in this.


Weakness:
- Evaluations primarily focus on novel rendering/relighting but lacks in individual intrinsic components such as 3D geometry (e.g. depth, chamfer, normal errors etc) and environment maps.
- Comparison with recent Monte Carlo inverse rendering method [31,37] especially [37] would strengthen this paper. (Although it should be noted [31] does not solve for geometry, while [37] assumes known lighting and does not have full code release).
- I think the paper is better suited for CV or graphics venues than NeurIPS. There are machine learning components in this paper but the main components (denoising/Monte Carlo rendering) do not seem very well aligned with NeurIPS.

---

> ### Author Response · Authors · 2022-08-01
> **1u76 individual questions**
>
> ## **Evaluations primarily focus on novel rendering/relighting but lacks in individual intrinsic components such as 3D geometry (e.g. depth, chamfer, normal errors etc) and environment maps**
>
> Please see the common section for additional evaluations.
>
> ## **Comparison with recent Monte Carlo inverse rendering method [31,37] especially [37] would strengthen this paper**
>
> As mentioned, [31] does not optimize geometry, and [37] assumes known lighting (a point light co-located with the camera).
> The visibility gradients of [31] require efficient silhouette detection, which is non-trivial to extend beyond
> primary visibility (requires complex 5D data structures for secondary rays, see e.g., "Unbiased Warped-Area Sampling for Differentiable Rendering" for a detailed discussion).
>
> Our core contribution is the joint optimization of shape, materials, and environment lighting using a Monte-Carlo renderer.
> To get a tractable optimization task (in terms of iteration times and noise levels), we restricted the renderer to direct illumination and added (differentiable) denoising to the pipeline.
> While there is currently no Monte Carlo path tracing system that tackles the same joint optimization task,
> the variance-reduction ideas introduced in this paper can be directly applied to future MC inverse rendering pipelines, once the computational resources are available.
>
> ## **I wonder to what extent the proposed method could be used to handle inter-reflections and light refraction in e.g. translucent objects**
>
> While the paper focuses on direct illumination, we discuss extensions to full path tracing briefly in the Conclusion section. It is a clear avenue for future work, but comes with additional challenges in
> increased noise-levels, visibility gradients through specular chains, and drastically increased iteration times. The denoising step is applicable as is. Please see common section for further details.

---

### Official Review · Reviewer_EEfj · 2022-07-11

**Rating:** 6
**Confidence:** 3
**Soundness:** 3 good
**Presentation:** 2 fair
**Contribution:** 2 fair

**Summary:**

The paper tackles the challenge of learning the shape, light & material properties of a scene from multiview images. The main idea is to consider a general lighting model for the rendering equation, which in turn is approximated using Monte Carlo (MC). As MC introduces large variance gradient estimates for optimization, the paper suggests importance sampling techniques. In addition, an image denoising model is incorporated into the system. The method is evaluated on standard scene datasets.

**Questions:**

Please address the weakness stated above.

**Limitations:**

The paper does not include such a discussion.

**Strengths And Weaknesses:**

Strengths
-------------
The paper is well written and easy to follow. The introduction is concisely informative.
The suggested system seems to improve previous work in learning scene properties decomposition.
The evaluation of synthetic data and real data is adequate.

Weaknesses
----------------

The paper focuses on the different solutions the proposed system consists of. However, I am missing a discussion on some of the challenges or design choices made in designing such a system. For example, the choice of the importance sampling techniques is given with any discussion. Some questions that arise are: Can the variance reduction be somehow quantified (bounded above)? Are there other existing techniques?
Another example of a missed discussion is how much the work is coupled with NVDIFFREC. Could the suggested techniques be incorporated into methods like [1] as well?

In addition, the discussion on the bias introduced (154-162) is unclear. The seems like an important issue that can also promote further research but is very shortly discussed.

The paper could be improved by being more self-contained. For example, the sampling techniques used are only referenced.

The effect of the new regularizer is not tested in an ablation study.

[1]: PhySG: Inverse Rendering with Spherical Gaussians for Physics-based Material Editing and Relighting, CVPR 2021.
[2]:

---

> ### Author Response · Authors · 2022-08-01
> **EEfj individual questions**
>
> ## **The choice of the importance sampling techniques is given with any discussion**
>
> The three importance sampling techniques we apply are well-proven in the computer graphics community, and applied in most modern production path tracers. Thus, we kept discussion very brief and cited the relevant methods.
> On request, we can add a more detailed summary of each sampling technique to the supplemental material to make the paper more self-contained.
> Applying these strategies in an inverse rendering pipeline is less explored. We use detached versions of these samplers.
> Recent work by Zeltner et al. [69] studies this problem in detail, and it is a nascent, active research field.
>
> ## **How much the work is coupled with NVDIFFREC. Could the suggested techniques be incorporated into methods like PhySG [1] as well?**
>
> Our denoising step works in image space and is directly compatible with any renderer. Similarly, our sampling works with any geometric representation supporting ray tracing (e.g. triangles, SDFs, ...).
> PhySG uses sphere tracing to evaluate the shape (SDF) and Spherical Gaussians to evaluate shading, w/o shadows.
> Replacing the PhySG renderer with a MC tracer was recently done in the paper "Differentiable Signed Distance Function Rendering" by Vicini et al.
> It drastically increases the noise levels compared to low-frequency SG representation of shading, which suggests that our sampling and denoising strategies could be helpful. Vicini et al. did not include results where the environment lighting is jointly optimized with shape and materials.
>
> ## **The discussion on the bias introduced (154-162) is unclear. The seems like an important issue that can also promote further research but is very shortly discussed**
>
> The variance-bias trade-off is indeed a very important topic. We purposely used biased rendering in our pipeline to get a tractable
> optimization problem. Note also that all denoisers inevitably introduce bias.
> We discuss it a bit further in the supplemental material (Fig 6, Section 4, Fig 10).
>
> ## **The effect of the new regularizer is not tested in an ablation study**
>
> Though not a full ablation, please refer to Fig 1 in supplemental material. We additionally show a full breakdown of the NeRF dataset in the newly added Figure 18 in supplemental material.
>
> Note that the regularizer is particularly well suited for scenes with more complex geometry and self-shadowing, such as Hotdog and Lego.
> The diffuse lighting encodes most of the shadows, while the Kd/Albedo term contains mostly chrominance, as expected.
> Our regularizer is less suited for scenes with simple geometry and complex materials. We still note that the albedo textures look somewhat improved,
> e.g., for the Materials scene. The Chair scene can be considered a failure case, as some material patterns are baked into lighting, still the material parameters look reasonable.
>
> Recent work on image delighting and shadow removal through neural networks is a promising alternative. While such methods will likely win in the long term,
> they did not perform well enough on our datasets in our initial experiments.

---

### Official Review · Reviewer_JQ7r · 2022-07-11

**Rating:** 8
**Confidence:** 4
**Soundness:** 4 excellent
**Presentation:** 4 excellent
**Contribution:** 3 good

**Summary:**

This paper presents an algorithm for the inverse rendering based on the neural fields. Unlike existing neural-fields-based inverse rendering relies on the simple shading model, the paper introduced the ray tracing with Monte Carlo integration that enables the inverse rendering more general and practical. To make the optimization tractable, the paper proposed to introduce multiple importance sampling and denoising for the differentiable ray tracing. The extensive evaluation shows that the proposed method can accurately recover physical attributes in more physically interpretable manner.

**Questions:**

- The paper claimed that the proposed method scales linearly with sample count but the Table at line 275 shows the iteration time doesn’t linear to the sample count. What does “scale linearly” mean?

- Can the proposed method be applied to the recovery of translucent or transparent objects with some realistic constraints (e.g., known background)?


**Limitations:**

The limitation is properly described.

**Strengths And Weaknesses:**

Strength:

- The introduction of this work is very clear which is supported by an extensive survey of previous studies.

- Utilizing the denoiser and importance sampling for stable and efficient differentiable rendering makes a lot of sense. I expect that this research will lead the computer vision community the active utilization of real-time rendering technics, which are being actively researched and developed in the field of computer graphics.

- Though most of the techniques for forward-rendering in this paper came from previous works, but their application to inverse rendering is not obvious. I think the beauty of this paper is that in every technical component, the reader can learn why it was introduced, including its surroundings thanks to the proper citations.

- The evaluation successfully demonstrated the benefit of the proposed denoising.

- The extensive ablation study justifies the algorithmic choice. It is really helpful to me to see the comparison of denoisers with detailed discussions in the supplementary material.

Weakness:

- As described in the paper, the method still considers the single bounce of light, therefore essentially inter-reflection is still a problem.

- Though qualitatively demonstrated, the reconstruction accuracy of individual attributes (such as normal or specular components) was not quantitatively evaluated.

- As the authors repeatedly mentioned, the training time is not as fast as expected even with the small sample count per pixel.

- Experiments show that the proposed method does a better job of recovering materials and lighting than the existing methods, but on the other hand, the increase in computational cost is an issue. Therefore, it is very helpful to clarify which kinds of scenes the existing method is not good at and the proposed method is good at handling so that a user can choose the proper method depending on the scene.

- The writing of the paper is very good, and the proposed method is undoubtedly very effective in recent inverse rendering based on Neural Fields. On the other hand, however, I had the impression that the paper uses existing techniques at a very high level rather than making innovative proposals, so I have slightly downgraded it from the highest evaluation.

---

> ### Author Response · Authors · 2022-08-01
> **JQ7r individual questions**
>
> ## **The reconstruction accuracy of individual attributes (such as normal or specular components) was not quantitatively evaluated**
>
> Please see the common section.
>
> ## **Therefore, it is very helpful to clarify which kinds of scenes the existing method is not good at and the proposed method is good at handling so that a user can choose the proper method depending on the scene**
>
> If material & light separation is important, we argue that this approach is preferable (by tracing shadow
> rays, we avoid shadows getting baked into the albedo texture). If view interpolation is the end goal, then NeRF does an excellent job.
>
> ## **What does “scale linearly” mean?**
>
> The computational cost of evaluating shading increases linearly with the number of samples (not counting denoising and geometry evaluation, which is a fixed cost, regardless of sample count). We use GPU ray tracing and measured the overall runtime of the system.
>
> ## **Can the proposed method be applied to the recovery of translucent or transparent objects with some realistic constraints (e.g., known background)?**
>
> It can likely be extended to support translucency and transparency, but we have left that for future work.

---

### Official Review · Reviewer_Hmte · 2022-07-13

**Rating:** 6
**Confidence:** 5
**Soundness:** 3 good
**Presentation:** 3 good
**Contribution:** 3 good

**Summary:**

The paper proposes a method to estimate the geometry, material and lighting
of an object from a set of multi-view images. It builds on top of previous work
nvdiffrec. Different from nvdiffrec that applies split-sum approximation for
direct lighting, the proposed method applies Monte Carlo ray tracing  and make the rendering more physically accurate. To reduce the
noise caused by Monte Carlo integration when
there are few samples, the paper proposed to use denoisers to reduce the
variance and make the training more efficient. The experiment results show that
the proposed method generates better results than baseline methods such as
nvdiffrec and nerfactor.


**Questions:**

See above.

**Limitations:**

The limitations are well discussed.

**Strengths And Weaknesses:**

Strengths:

The paper combines neural inverse rendering and traditional Monte Carlo ray
tracing, which enjoys the benefits of both world.
The idea of using denoisers to reduce variance in Monte Carlo integration
is interesting and helps make the training more efficient. The paper did
multiple experiments to show that it helps improve material, geometry and
lighting estimation.

Weakness/Questions:
1. As mentioned in the paper, while using Monte Carlo integration, the paper
didn't consider multi-bounces or shadow gradients, which makes the method fail
to handle indirect illumination. As a stress test, it would be interesting to
see how it works when increasing the number of bounces.

2. For shadow gradients, how the light leakage term is set? Is it a fixed
parameter for all experiments?

3. In Section 3 of the supp, it's not clear to me how the visibility gradients
are calculated?  Is the paper using the methods based on warped area/edge sampling?
It's worth adding a more detailed discussion here to differentiate the referred
methods and the method used by the paper.

4. How robust is the denoiser in the optimization? At the early stage of the
optimization, when the rendering images don't make too much sense, will denoiser
make the optimization worse? The paper talks about ramping up the spatial
footprint and linearly blending the noisy and denoised images in supp (Line 60),
how robust are those operations? Do they need to be fine-tuned for each scene?

5. During optimization, since it minimizes the difference between the denoised
images and the GT images, I am wondering whether it's possible that the noisy
images have some artifacts that are not caused by low samples but incorrect
estimations. Such artifacts will be covered by the denoiser, and will be visible
during testing. Is such a case possible? During testing, can we get high-quality
noise-free results by increasing the number of samples without using denoisers?


6. In Line 266, the paper says view interpolation degrades with improved
material and lighting estimations. More materials/evidence are needed to support this
claim.

Overall, I think the idea of combining neural inverse rendering, traditional
Monte Carlo integration and denoisers is interesting and novel. The paper did
thorough evaluations on the method and performs detailed ablation study.
Therefore, I vote for accepting the paper.

---

> ### Author Response · Authors · 2022-08-01
> **Hmte individual questions**
>
> ## **Q1: Multi-bounce**
> ​
> Please see the common section.
>
> ## **Q 2 & 4: Light leakage / How robust is the denoiser in early optimization**
> ​
> Note that the light leakage term is not strongly related to shadow (silhouettes/boundary) gradients, but is rather a trick to avoid gradient discontinuities in early training, similar to the denoiser footprint schedule.
>
> The images indeed do not make much sense in early training, and large filters / strong shadows can cause geometry optimization (in particular) to get stuck in a local minima.
> E.g., optimization may fail to carve out geometry, because a strong shadow will temporarily be added until neighboring geometry is removed.
> We schedule light leakage and filter footprint in tandem, with a simple linear ramp over the first 1750 iterations of the first optimization pass:\
> $\mathrm{light\_leak} = \max(0, 1 - i / 1750)$, \
> $\sigma = \sigma_{\mathrm{max}} * \min(1, i / 1750)$, \
> where $i$ is the iteration number. We used the *exact same schedule* for all scenes in the paper.
> Given the large diversity of geometric complexity in our scenes, we consider it robust.
>
> ## **Q3: Shadow gradients**
> ​
> Note that we did not use visibility gradients for shadow rays in the main paper, as we didn't see a clear benefit in our multi-view setting (as discussed in Section 3.1). For the ablation in the supplemental, we used an extension of the nvdiffrast AA test, extended to 3D, to compute shadow gradients, and we expect similar results using other recent gradients options (warped area/edge sampling). A limited evaluation with visibility gradients based on Warped-Area Sampling indicates the same behavior.
> ​
> ## **Q5: During testing, can we get high-quality noise-free results by increasing the number of samples without using denoisers?**
> ​
> Please see the common section.
>
> ## **Q6: The paper says view interpolation degrades with improved material and lighting estimations. More materials/evidence are needed to support this claim**
> ​
> Our claim is too strong and we will reformulate it. We enforce material/light separation through additional regularization. Without the regularizer loss terms we would optimize for the same (PSNR) image loss used for validation, which should yield a better view interpolation result assuming optimization finds the global minimum. Compared to NeRF, all previous work which provides material separation (NeRD, NeRFactor, nvdiffrec, PhySG) reduces quality, but there is no strict correlation between improved material separation and decreased view interpolation quality.

---

### Author Response · Authors · 2022-08-01
**Common section**

We thank all the reviewers for the great feedback. In this section, we address some common concerns. Please also refer to the individual responses. To help other researchers reproduce our results, we intend to release source code upon acceptance.

## **Multi-bounce results**

Our paper presents results on direct illumination (environment lighting with shadows). As discussed in our conclusions, multi-bounce
path tracing is a clear (and exciting) avenue for future work, but comes with additional challenges in increased noise-levels, visibility gradients through specular chains, and drastically increased iteration times.

We leverage importance sampling and denoising techniques from production path tracers, and argue that
these techniques will be important building blocks for future high quality inverse rendering pipelines,
e.g., the denoising step is applicable as is.
Our approach is a small step towards multi-bounce MC inverse rendering pipelines.

Furthermore, neural materials pose an engineering challenge. To make optimization tractable in terms of memory consumption, we need to apply "path replay",
(see "Path Replay Backpropagation: Differentiating Light Paths using Constant Memory and Linear Time" by Vicini et al. for details) which require fusing the path tracing and MLP kernels, which is non-trivial in the tinycudann system we currently apply for neural material evaluation.

## **Additional terms in evaluation, e.g. depth, chamfer, normal errors, environment maps**

We present additional quantitative results below for individual terms. Please also refer to Figure 8 in the paper, where we show a visual breakdown of material parameters and normals compared to nvdiffrec for three scenes. We also added visual results to the supplemental material (Fig 17-18)
to show the geometry quality and regularizer impact.

### **Albedo texture**

As noted by NeRFactor & nvdiffrec, there is an indeterminable scaling constant per (R, G, B) value between texture albedo and the environment light (bright material and dim light or vice versa), which makes quantitative evaluation of textured albedo quality challenging.
This constant can often completely dominate the error. To sidestep this issue, following NeRFactor and nvdiffrec, in the table below we normalize albedo by the average intensity of the reference albedo.

NeRFactor dataset:
| PSNR (dB)       |  Drums |  Ficus | Hotdog |   Lego |
|-----------------|--------|--------|--------|--------|
| nvdiffrec       |   20.7 |   31.6 |   22.8 |   19.1 |
| Our             |   20.9 |   32.8 |   22.2 |   22.3 |

NeRF dataset:
| PSNR (dB)       |  Chair | Hotdog |   Lego |    Mic |
|-----------------|--------|--------|--------|--------|
| nvdiffrec       |   24.2 |   18.9 |   19.0 |   28.3 |
| Our             |   25.5 |   18.3 |   21.2 |   27.1 |

The aforementioned scaling factors make traditional image metrics a poor measure of material quality.
In the newly added Figure 18 in supplemental material we show a material component breakdown for the NeRF scenes.
While the PSNR albedo scores for the Hotdog scene are relatively similar, we argue that our desaturated albedo would
be easier for an artist to cleanup/edit.

### **Normal**

We evaluate normal quality in image space based on the rendered normal g-buffer (as shown in Figure 8). For the NeRFactor dataset
we note minor differences in normal quality compared to nvdiffrec.

NeRFactor dataset
| PSNR (dB)       |  Drums |  Ficus | Hotdog |   Lego |
|-----------------|--------|--------|--------|--------|
| NVDIFFREC       |   20.0 |   24.8 |   20.9 |   14.5 |
| Our             |   19.9 |   24.6 |   20.8 |   15.0 |

### **Chamfer loss**

In the table below we relate our method to Table 8 (supplemental) of the nvdiffrec paper, and note similar geometric quality as nvdiffrec.
The Lego scene is an outlier caused by our normal smoothness regularizer. The scene is particularly challenging for geometric smoothing
since it contains very complex geometry full of holes and sharp edges.

|                 |  Chair | Hotdog |   Lego |  Mats. |   Mic |
|-----------------|--------|--------|--------|--------|-------|
| PhySG           | 0.1341 | 0.2420 | 0.2592 |    N/A | 0.2712|
| NeRF (w/o mask) | 0.0185 | 4.6010 | 0.0184 | 0.0057 | 0.0124|
| NeRF (w/ mask)  | 0.0435 | 0.0436 | 0.0201 | 0.0082 | 0.0122|
| nvdiffrec       | 0.0574 | 0.0272 | 0.0267 | 0.0180 | 0.0098|
| Our             | 0.0566 | 0.0297 | 0.0583 | 0.0162 | 0.0151|

Please refer to Figure 17 in the updated supplemental work for a visual comparison.

## **During testing, can we get high-quality noise-free results by increasing the number of samples without using denoisers?**
​
We will update the paper to be more clear. At test time, all view interpolation results are generated **without** denoising at high sample counts. All relighting results are rendered in Blender with moderate-to-high sample counts and using Blenders denoising algorithm (which is different from ours).

---

### Meta-Review · Area_Chair_Kibw · 2022-08-23

**Recommendation:** Accept
**Confidence:** Certain

**Metareview:**

The paper addresses the task of reconstructing 3D meshes, materials and lighting from multi-view images. To this end Monte Carlo ray tracing is used in combination with denoisers for training efficiency. Experiments show improvement w.r.t. SOTA nvdiffrec and nerfactor.
Reviewers like the combination of denoisers for training and MC rays tracing, the thorough evaluation and ablation studies. All reviewers recommend the paper for acceptance and so do I.

**Award:**

No

---

### Decision · Program_Chairs · 2022-09-14

Accept